# HAM/TSP Pathogenesis: The Transmigration Activity of HTLV-1-Infected T Cells into Tissues

**DOI:** 10.3390/pathogens12030492

**Published:** 2023-03-21

**Authors:** Tatsufumi Nakamura

**Affiliations:** Department of Social Work, Faculty of Human and Social Studies, Nagasaki International University, Nagasaki 859-3298, Japan; tatsu@niu.ac.jp or tnakamura161226@gmail.com; Tel.: +81-956-20-5508

**Keywords:** HTLV-1, HAM/TSP, chronic myelitis, transmigration, adhesion molecule, integrin, small GTPase, matrix metalloproteinases

## Abstract

Slowly progressive spastic paraparesis with bladder dysfunction, the main clinical feature of human T-cell leukemia virus-1 (HTLV-1)-associated myelopathy/tropical spastic paraparesis (HAM/TSP), is induced by chronic inflammation in the spinal cord, mainly the lower thoracic cord. A long-standing bystander mechanism, such as the destruction of surrounding tissues by inflammatory cytokines, etc., induced under the interaction between infiltrated HTLV-1-infected CD4^+^ T cells and HTLV-1-specific CD8^+^ cytotoxic T cells, has been considered implicated for the induction of chronic inflammation. As this bystander mechanism is triggered conceivably by the transmigration of HTLV-1-infected CD4^+^ T cells to the spinal cord, heightened transmigrating activity of HTLV-1-infected CD4^+^ T cells to the spinal cord might play a crucial role as the first responder in the development of HAM/TSP. This review evaluated the functions of HTLV-1-infected CD4^+^ T cells in HAM/TSP patients as the prerequisite for the acquisition of the activity such as adhesion molecule expression changes, small GTPases activation, and expression of mediators involved in basement membrane disruption. The findings suggest that HTLV-1-infected CD4^+^ T cells in HAM/TSP patients have enough potential to facilitate transmigration into the tissues. Future HAM/TSP research should clarify the molecular mechanisms leading to the establishment of HTLV-1-infected CD4^+^ T cells as the first responder in HAM/TSP patients. In addition, a regimen with an inhibitory activity against the transmigration of HTLV-1-infected CD4^+^ T cells into the spinal cord might be recommended as one of the therapeutic strategies against HAM/TSP patients.

## 1. Introduction

Human T-cell leukemia virus-1 (HTLV-1)-associated myelopathy/tropical spastic paraparesis (HAM/TSP) is a chronic progressive myelopathy characterized by bilateral pyramidal tract involvement with sphincteric disturbances [1,2]. Although the exact reasons why HTLV-1 induces HAM/TSP in a very small population, such as 0.3–3.8% of HTLV-1-infected individuals [3,4], are still unsolved, the involvement of numerous immunological dysregulations based on high HTLV-1 proviral load in the peripheral blood mononuclear cells (PBMC) are strongly suggested in the development of HAM/TSP [5,6,7]. Although the abundance of interferon-γ (IFN-γ) and tumor necrosis factor-α(TNF-α) producing cells in the HTLV-1 tax-expressing cell population in PBMC of HAM/TSP patients was observed in a comparative study of intracellular cytokine expression levels in HAM/TSP patients and HTLV-1 asymptomatic carriers with a high HTLV-1 proviral load equivalent to those of HAM/TSP patients [8], this finding suggested that HTLV-1-infected cells having the characteristic of Th1 increased in HAM/TSP patients. Araya et al. clearly demonstrated that IFN-γ-producing CD4^+^CCR4^+^ T cells expressing Th1 marker CXCR3 based on the activation of the Th1 master regulator T box transcription factor (T-bet) induced by HTLV-1 tax in cooperation with specificity protein 1 increased in cerebrospinal fluid (CSF) and spinal cord lesions of HAM/TSP patients [9]. In addition, the frequency of this cell population in PBMC was correlated with disease activity of HAM/TSP [10]. Thus, a Th1-like status based on a high HTLV-I proviral load in the peripheral blood may be very critical in the immunopathogenesis of HAM/TSP. The primary neuropathological feature of HAM/TSP is chronic myelitis, such as chronic inflammation in the spinal cord, mainly the lower thoracic cord, with perivascular cuffing and parenchymal infiltration of mononuclear cells [11]. The exact mechanism of how HTLV-1 infection causes chronic inflammation of the spinal cord is still unclear. However, a long-standing bystander mechanism, such as the destruction of surrounding tissues by inflammatory cytokines, etc., induced under the interaction between HTLV-1-infected Th1-like CD4^+^ T cells and HTLV-1-specific CD8^+^ cytotoxic T cells in the spinal cord, is probably a critical cause of the induction of chronic myelitis [2,12,13] (Figure 1). Although there is no significant difference in a functional CD8^+^ cell assay for the anti-viral efficacy of HTLV-1-specific CD8^+^ cytotoxic T cells between HAM/TSP patients and HTLV-1 carriers [14], the finding that HTLV-1-specific CD8^+^ cytotoxic T cells of HAM/TSP patients as the response to HTLV-1 tax produces high levels of proinflammatory cytokines, such as IFN-γ and TNF-α, etc., [15] strongly suggests that HTLV-1-specific CD8^+^ cytotoxic T cells can function for the induction of bystander mechanism. In addition to this situation, a positive feedback loop, which is formed by chemokine CXCL10 induced from astrocytes via stimulation by IFN-γ produced by infiltrated HTLV-1-infected Th1-like CD4^+^ T cells, might be involved in the maintenance and promotion of chronic myelitis [16]. Furthermore, non-HTLV-1-infected CXCR3^+^ cells, which are attracted by CXCL10, provide the additive effect for a positive feedback loop (Figure 1).

As this bystander mechanism is triggered conceivably by the transmigration of HTLV-1-infected CD4^+^ T cells to the spinal cord, heightened transmigrating activity of HTLV-1-infected CD4^+^ T cells to the spinal cord might play a crucial role in the first stage of the development of HAM/TSP. The fact that HTLV-1 provirus accumulated in the CSF in HAM/TSP patients compared to HTLV-1 asymptomatic carriers [17,18] and that the main lesion of the spinal cord in HAM/TSP is in the lower thoracic cord (the watershed zone providing stagnant blood flow in hemodynamic condition) [19,20] might indicate that the outcome is induced by heightened transmigrating activity of HTLV-1-infected CD4^+^ T cells of HAM/TSP patients. Several inflammatory diseases, including uveitis, pulmonary disorder, arthritis, myositis, and Sjögren syndrome, occasionally occur in conjunction with HAM/TSP [21]. Even in this situation, heightened transmigrating activity of HTLV-1-infected CD4^+^ T cells to the tissues might be involved in the triggering of these inflammatory diseases.

However, the exact mechanisms of how HTLV-1-infected CD4^+^ T cells acquire this function as the first responders in the pathogenesis of HAM/TSP, including HAM/TSP-related inflammatory diseases, still remain unresolved. In this review, the focus will be on the findings relevant to the heightened transmigration activity of HTLV-1-infected CD4^+^ T cells into the tissues involved in the pathogenesis of HAM/TSP.

## 2. The Change of Cell Adhesion-Related Molecules Expression on HTLV-1-Infected CD4^+^ T Cells Leading to the Heightened Transmigrating Activity through Vascular Endothelial Cells in HAM/TSP Patients

Studies have demonstrated that the expression of several kinds of adhesion molecules, such as intercellular adhesion molecule-1 (ICAM-1) and vascular cell adhesion molecule-1 (VCAM-1), is upregulated in HTLV-1-infected cells [22].

In the process leading to the transmigration of T cells into the tissues, the rolling process of T cells on vascular endothelial cells (ECs) functions as the first step [23]. Selectin and its ligands, which are expressed on ECs and T cells, respectively, play an important role in this step [24,25]. Of these, sialyl Lewis^x^ antigen (sLe^x^) is a ligand for both E- and P-selectin [26]. Therefore, T cells expressing sLe^x^ (sLe^x+^ cell) might have the potential to transmigrate into tissues. We previously compared the frequency of sLe^x+^ cells together with IFN-γ production in peripheral blood CD4^+^ T cells between 8 HAM/TSP patients and 14 controls, including four anti-HTLV-1-seropositive carriers [27]. As shown in the results, the frequency of sLe^x+^ cells in peripheral blood CD4^+^ T cells of HAM/TSP patients was significantly higher than in controls. In addition, the activity of IFN-γ production in the sLe^x+^ cell population in the peripheral blood CD4^+^ T cells of HAM/TSP patients had significantly increased compared to controls. Futhermore, when we compared HTLV-1 proviral load between sLe^x+^ and sLe^x−^ cell populations, HTLV-1 proviral load in the sLe^x+^ cell population was two- to eight-fold higher than the sLe^x−^ cell population. Thus, in the peripheral blood CD4^+^ T cells of HAM/TSP patients, the frequency of sLe^x+^ cells (in which Th1 cytokine, such as IFN-γ, production is activated) increased, and the HTLV-I provirus was concentrated in the sLe^x+^ cell population. These findings suggested that HTLV-1-infected Th1 cells having the potential to cause the trigger for transmigration to the spinal cords are increased in the peripheral blood CD4^+^ T cells of HAM/TSP patients.

Following the rolling process on ECs, the firm adhesion of T cells by the interaction between integrins and their ligands plays a crucial role as the preliminary step for transmigration of T cells into the tissue [25,28].

We previously reported significantly increased adherence of peripheral blood T cells to ECs in HAM/TSP patients [29] as evidenced by the adherence to and the transmigration through ECs of activated CD4^+^ T cells, but not CD8^+^ T cells, with heightened lymphocyte function antigen-1 (LFA-1) expression belonging to integrin family [30,31]. Romero et al. demonstrated that the enhanced adhesion of HTLV-1-infected cells to and the transmigration of HTLV-1-infected cells through the brain ECs is followed by increased brain endothelial permeability along with increased expression of LFA-1 [32]. This suggests that the enhanced adhesion activity of HTLV-1-infected cells to ECs itself induces not only the increased transmigration of HTLV-1-infected cells into the tissues but also the facilitation of the disruption of the blood–brain barrier (BBB), leading to the influx of harmful agents to the tissues. The disruption of BBB, such as the change in tight junctions between Ecs, was observed in the spinal cord of HAM/TSP patients [33]. This BBB breakdown, which is supposed to be induced by HTLV-1 infection to ECs [34] or proinflammatory cytokines secretion [33] following the increased adhesion to ECs of HTLV-1-infected cells, might strongly contribute to the acceleration of the neuronal damages induced by the bystander mechanism. In addition, Curis et al. reported in the study of in vitro BBB model that the overexpression of activated leukocyte cell adhesion molecule (ALCAM/CD166), which is a member of the immunoglobulin superfamily, on HTLV-1-infected cells induced their increased trafficking through ECs [35]. In this study, they also showed that the downregulation of ALCAM expression significantly reduced the migration of HTLV-1-infected cells derived from HAM/TSP patients but not from healthy donors, although the change of ALCAM expression was not observed in the BBB ECs in the staining of spinal cord sections of HAM/TSP patients. These findings suggest that ALCAM overexpression on HTLV-1-infected cells is involved in the transmigration of HTLV-1-infected cells to the spinal cord in HAM/TSP patients. There have been reports of overexpression of cell adhesion molecule 1/tumor suppressor in lung cancer 1 (CADM1/TSLC1), a member of the immunoglobulin superfamily, in HTLV-1-infected cells and adult T cell leukemia (ATL) cells [36,37]. Recently it was reported that the expression of this adhesion molecule is higher in CD4-positive T cells of HAM/TSP patients compared to the expression in HTLV-1 asymptomatic carriers [38].

On the other hand, the abnormalities of very late antigen-4 (VLA-4) (a member of the integrin family and a ligand of vascular cell adhesion molecule-1(VCAM-1), expressed on T cells) are also reported. Immunopathological analysis of spinal cord lesions in HAM/TSP patients revealed that infiltrating mononuclear cells expressed VLA-4 concomitantly with highly expressed VCAM-1 on the vascular endothelium were detectable on perivascular infiltrating cells and endothelium in active chronic lesions [39]. Thus, the interaction between VLA-4 and VAM-1 also seems to be important in the pathogenesis in the spinal cord of HAM/TSP patients. Recently, there has been reported increase of CD49d, which is a constituent of VLA4, positive cells in peripheral blood T cells of HTLV-1-infected patients, including HAM/TSP patients [40].

Thus, the upregulation of cell adhesion molecule expression on HTLV-1-infected cells of HAM/TSP patients might have enough potential to facilitate the transmigration of HTLV-1-infected cells through ECs. However, although the expressions or the activities of LFA-1, LFA-3, ICAM-1, VCAM-1, and ALCAM, etc., are associated with HTLV-1 tax or HBZ activity [22,35,41], the effects of selective adhesion molecules on HTLV-1-infected T cells of HAM/TSP have not been clarified yet. Therefore, when considering the potential therapeutic target for HAM/TSP patients, the identification of a specific adhesion molecule might have an important significance in the future.

## 3. The Role of Small GTPase Activation Leading to the Heightened Transmigration Activity of HTLV-1-Infected CD4^+^ T Cells into Tissues in HAM/TSP Patients

Small GTPases function as the propulsion for the transmigration of the cells through ECs. That is, small GTPase activation can induce the upregulation of transmigration activity of the cells into tissues through the rearrangement of the cytoskeleton involved in the adhesion and migration of cells [42]. The finding that HTLV-1 tax regulates cell adhesion and migration by the interaction with small GTPases, such as Cdc42, Rac, and Rho, was previously demonstrated [43]. A report indicated that Gem, which is one of the small GTP-binding proteins belonging to the Ras superfamily, is involved in the increase of the cellular migration of HTLV-1-infected cells through cytoskeleton remodeling [44].

We previously reported the significantly increased adherence to ECs and subsequent transmigration through ECs of activated CD4^+^ T cells with heightened LFA-1 expression in the peripheral blood T cells of HAM/TSP patients as mentioned above [30,31], suggesting the upregulation of signaling based on integrin/ligand interaction in the peripheral blood CD4^+^ T cells of HAM/TSP patients. Subsequently, the upregulation of integrin/ligand signaling induces the activation of small GTPases, which are the downstream targets, followed by the rearrangement of cytoskeletal components [45,46]. Therefore, small GTPases might be activated in the HTLV-1-infected cells of HAM/TSP patients. To confirm this, we analyzed the activity of small GTPases, such as Cdc42, Rac, and Rho, in HTLV-1-infected T cell lines derived from HAM/TSP patients in comparison with those in HTLV-1-infected T cell lines derived from other origins [47]. As a result, it was revealed that all small GTPases were strongly activated in all cell lines derived from HAM/TSP patients. Of these small GTPases, the difference in the degree of activation between both kinds of cell lines was the most obvious in Cdc42. Cdc42 plays an important role in the polarization of the cytoskeleton following integrin-mediated activation [48] with the involvement in cell migration [49,50]. Therefore, the activation of Cdc42 in the HTLV-1-infected cells of HAM/TSP patients suggests that these cells have upregulated transmigrating activity into the tissues. Thus, activation of the outside-in signaling from integrin signaling in HTLV-1-infected cells of HAM/TSP patients suggests that this activity functions as one of the first triggers in the development of HAM/TSP.

Integrins can also conduct inside-out signaling, and this function also might be involved in transmigrating activity into the tissues of HTLV-1-infected cells of HAM/TSP patients. Rap1, a small GTPase that functions in a complex with adaptor molecules, such as RAPL, talin1, and kindlin-3, facilitates the inside-out signaling for integrin signaling, leading to increased adhesion to integrin ligands [51,52]. Activation of Rap1 through C-X-C chemokine receptor type 4 signaling was observed in the analysis of an HTLV-1-infected T cell line derived from a HAM/TSP patient [53].

Thus, activation of small GTPases through both outside-in signaling from integrin and inside-out signaling for integrin in HTLV-1-infected cells of HAM/TSP patients can induce heightened transmigrating activity into the tissues, and these activities might function as one of the first responder roles in the development of HAM/TSP [54] (Figure 2).

## 4. The Mediators Involved in the Heightened Tissue Transmigration of HTLV-1-Infected CD4^+^ T Cells in HAM/TSP Patients

When considering the transmigration of T cells into the tissues after passing through the endothelium barrier, the extracellular matrix, including the vascular basement membrane, functions as the next barrier. Although collagens, gelatine, fibronectin, and laminin are the main components of the vascular basement membrane, matrix metalloproteinases (MMPs), such as MMP-2 and MMP-9, can cleave these components followed by the disruption of the basement membrane of the endothelium [55]. Indeed, the immunopathological analysis of spinal cord lesions in HAM/TSP patients revealed that MMP-2 and MMP-9 are expressed in infiltrating mononuclear cells with disruption of the vascular endothelium in chronic active lesions with the findings that higher levels of MMP-2 and/or MMP-9 were detected in the CSF of HAM/TSP patients [56,57]. In addition, the importance of MMPs in the tissue transmigration of T cells by the degradation of the extracellular matrix is also supported by the finding that the transmigration of CD4^+^ T cells of HAM/TSP patients was significantly inhibited by selective MMP inhibitor [58]. Recently, the upregulation of TNF-α-induced production of MMP-9 in PBMC culture with a higher MMP-9/TIMP-3 (an inhibitor of MMP-9) ratio in HAM/TSP patients was reported, although it is unclear which cell population in PBMC is involved in the increased production of MMP-9 [59].

Recent proteomic analysis of CSF revealed an increased level of soluble VCAM-1 (sVCAM-1) in HAM/TSP patients [60,61]. It is reported that the production of MMPs under the inflammatory status induces the shedding of sVCAM-1 from the surface of ECs [62,63]. Therefore, the up-regulated expression of MMPs in CSF, as mentioned above, might account for the increase of sVCAM-1 in CSF of HAM/TSP patients.

When considering the importance of MMP expression in the contact to cell adhesion molecules, we previously demonstrated that VCAM-1-mediated MMP-2 induction was up-regulated in T cells of HAM/TSP patients [64]. This finding might be in support of the possibility that VCAM-1/VLA-4 interaction is one of the important ingredients that induces the trigger of the transmigration of HTLV-1-infected cells to the spinal cord in HAM/TSP patients.

To evaluate the transmigrating activity of peripheral blood CD4^+^ T cells of HAM/TSP patients through the basement membrane, we previously investigated the transmigration of peripheral blood CD4^+^ T cells of HAM/TSP patients through the reconstituted basement membrane (RBM) using Transwell inserts, which were polyvinylpyrrolidone-free polycarbonate filters pre-coated with laminin on the lower surface and RBM (Matrigel) on the upper surface, respectively [65]. The results showed that the transmigrating activity of peripheral blood CD4^+^ T cells of HAM/TSP patients through RBM was significantly higher than that of either HTLV-1-seropositive carriers or HTLV-1-seronegative controls. Subsequently, we found that HTLV-1 proviral load in transmigrated CD4^+^ T cells from HAM/TSP patients was significantly higher than in non-transmigrated CD4^+^ T cells. By contrast, no significant difference was found in HTLV-1 proviral load in transmigrated and non-transmigrated CD4^+^ T cells from HTLV-1-seropositive carriers. These results indicated that peripheral blood CD4^+^ T cells of HAM/TSP patients, particularly HTLV-1-infected CD4^+^ T cells, have heightened transmigrating activity based on the induction of RBM disruption, suggesting that the character of the heightened transmigrating activity of HTLV-1-infected CD4^+^ T cells through the vascular basement membrane acquires the ability to accumulate in the tissues and is consistent with the findings that HTLV-1 provirus accumulates in CSF in HAM/TSP patients compared to HTLV-1 asymptomatic carriers as earlier described [17,18]. As the mediators involved in the disruption of RBM by HTLV-1-infected CD4^+^ T cells of HAM/TSP patients, the increased activity of aminopeptidase-N (APN), which is one of cell surface proteases as same as MMPs [66] was observed.

Thus, these findings strongly suggest that HTLV-1-infected CD4^+^ T cells in HAM/TSP patients have sufficient activity to induce mediators to disrupt the vascular basement membrane for tissue transmigration.

Finally, the series of molecular events involved in the heightened transmigration activity of HTLV-1-infected CD4^+^ T cells in HAM/TSP patients into tissues, as mentioned above, is summarized in Figure 2.

## 5. Other Inflammatory Diseases in HAM/TSP Patients

Several inflammatory diseases, such as Sjögren’s syndrome [67], myositis [68], uveitis [69], pulmonary disorder [70], arthritis [71], and infective dermatitis [72], occasionally occur in HTLV-1-infected individuals, including HAM/TSP patients [21]. Although it is necessary to consider the pathogenesis in those complicated with HAM/TSP in distinction from those complicated with HTLV-1 carrier status, at least, the exact pathogenesis of these inflammatory diseases is still obscure.

We previously reported interstitial cystitis and persistent prostatitis in HAM/TSP patients with concomitant appearance of anti-HTLV-1 antibody in urine [73] and HAM/TSP patients with multi-organ inflammatory disease, including Sjögren’s syndrome, uveitis, and interstitial cystitis, in which the existence of HTLV-1 provirus was confirmed in the salivary gland, aqueous humor, and mucosa of the urinary bladder with high HTLV-1 proviral load in PBMC [74]. The case of a HAM/TSP patient with Sjögren’s syndrome and lymphocytic pneumonitis was also reported [75].

We previously reported that Sjögren syndrome occurs very frequently in HAM/TSP patients [76]. In this study, definitive Sjögren syndrome was diagnosed in six out of ten patients. However, in the immunohistological studies of T-cell infiltration, predominantly CD3^+^CD4^+^ cells, was observed in the labial salivary glands of all examined HAM/TSP patients, suggesting that CD4^+^ T cells of HAM/TSP patients have transmigrating activity into the salivary glands even if its occurrence is not severe enough to induce the pathological changes that cause Sjögren syndrome, although the determination of these CD4^+^ T cells as HTLV-1-infected cells was not investigated. In addition, we analyzed the existence of ectopic germinal centers (GCs), one of the characteristic pathological features of Sjögren syndrome [77], in the labial salivary glands. Ectopic GCs were not detected in HAM/TSP patients with Sjögren syndrome [78], suggesting that Sjögren syndrome with HAM/TSP is not based on GC formation but might be based on CD4^+^ T cell infiltration into the salivary glands.

Overall, the HTLV-1-infected T cells in HAM/TSP patients might also have the potential to induce the trigger for the development of systemic inflammatory status as the first responder or maintain its status by the heightened transmigrating activity into the tissues.

## 6. The Polysulfate Treatment for HAM/TSP Patients Focusing on the Inhibition of the Transmigration of HTLV-1-Infected Cells into the Spinal Cord

The ideal treatment for HAM/TSP patients is the complete elimination of HTLV-1-infected cells. However, the therapeutic strategy against HAM/TSP is yet to be established. Therefore, a regimen with an inhibitory activity against the transmigration of HTLV-1-infected cells as the first responder into the spinal cord might be recommended as one of the therapeutic strategies. The main regions in which pathological changes occur in HAM/TSP are in the lower thoracic spinal cord [11]. These regions are anatomical watershed zones [19,20], where lymphocytes stagnate because of the decreased blood flow, can easily transmigrate to the tissues and evoke immune reactions. Therefore, manipulation of the microcirculation and interaction between lymphocyte and vascular ECs might be one of the therapeutic strategies against HAM/TSP.

Pentosan polysulfate sodium (PPS), which was developed as a heparin-like agent and has been used in Europe for thrombosis prophylaxis and treatment of phlebitis, is a semisynthetic drug manufactured from European beech-wood hemicellulose by sulfate esterification [79]. Therefore, PPS is safe and has also been approved by the US Food and Drug Administration as an oral medication for treating interstitial cystitis. In addition to the activity of improving microcirculation, polysulfates, such as heparin and PPS, have the potential to inhibit the intercellular spread of HTLV-1 by blocking the binding of the virus to heparan sulfate proteoglycans [80,81]. Indeed, the multiple activities of PPS experimentally verified included (i) the inhibition of the adhesion to and transmigration of HTLV-1-infected cells through ECs; (ii) inhibition of HTLV-1 cell to cell transmission; (iii) suppression of HTLV-1 production; and (iv) blockage of interaction of HTLV-1 infection and ECs with the inhibition of subsequent induction of inflammatory cytokines [82].

Based on the efficacy of heparin treatment, as demonstrated in our previous clinical trial against HAM/TSP patients [83], we tested the effect of the administration of PPS for 8 weeks in 12 patients with HAM/TSP in an open-labeled design [84]. As a result, this treatment induced marked improvement in lower-extremity motor function based on reduced spasticity, such as a reduced time required for walking 10 m. In the study, there were no significant changes in HTLV-1 proviral load in the peripheral blood, contrary to the inhibitory effect of PPS in vitro for the intercellular spread of HTLV-1. However, serum sVCAM-1 was significantly increased with a positive correlation to clinical improvements, such as a reduced time required for walking 10 m. Although the mechanisms of how serum sVCAM-1 is increased by PPS treatment is unclear, PPS might induce neurological improvement by the attenuation of the positive feedback loop of chronic inflammation in the spinal cord [16] through blocking of the adhesion cascade by increasing serum sVCAM-1 in addition to the rheological improvement of the microcirculation. PPS has enough potential to be a new therapeutic strategy for HAM/TSP.

## 7. Conclusions and Perspectives

Considering the mechanism of chronic inflammation induced in the spinal cord of HAM/TSP patients, HTLV-1-infected CD4^+^ T cells might play a crucial role as the first responder. Therefore, the activities of HTLV-1-infected CD4^+^ T cells in HAM/TSP patients were evaluated based on the transmigration activity into the tissues. Consequently, it was demonstrated that HTLV-1-infected CD4^+^ T cells in HAM/TSP were supposed to have enough heightened transmigrating activity into the tissues based on the activation of integrin signaling followed by small GTPase activation with the up-regulated expression of MMPs. This activity appears sufficient to allow HTLV-1-infected CD4^+^ T cells to function as the first responders in the development of HAM/TSP. In addition, the HTLV-1-infected CD4^+^ T cells in HAM/TSP patients might also have the potential to induce the trigger for the development of another systemic inflammatory status, including Sjögren’s syndrome, myositis, and uveitis, etc., which occasionally occur in conjunction with HAM/TSP, as the first responder.

Although the exact reasons why HTLV-1 induces HAM/TSP in a very small population of HTLV-1-infected individuals are still unsolved, a very important task in future HAM/TSP research is to clarify the molecular mechanisms leading to the establishment of HTLV-1-infected CD4^+^ T cells as the first responder in HAM/TSP patients. At this point, the acquisition of heightened transmigrating activity of HTLV-1-infected CD4^+^ T cells into the tissues among HTLV-1-infected individuals seems to be the key event leading to the development of HAM/TSP. However, it is still unclear how the change of signal molecules supporting the expression of the heightened transmigrating activity of HTLV-1-infected CD4^+^ T cells in HAM/TSP patients demonstrated in this review are induced among HTLV-1-infected individuals. For example, is it based on the existing signal molecules or other novel signal molecules, or is it simply based on the increase of HTLV-1-infected Th1 cells?

In small GTPase activation, although the finding that HTLV-1 tax regulates cell adhesion and migration by the interaction with Cdc42, Rac, and Rho involved in the outside-in signaling from integrin signaling was previously demonstrated [43], the interaction between HTLV-1 and Rap1 involved in the inside-out signaling for integrin signaling is still unclear. Therefore, the clarification of the signaling pathway or signaling molecule relevant to Rap1 activation might contribute to the elucidation of the molecular mechanism for HAM/TSP development.

There are currently no therapeutic strategies, such as the elimination of HTLV-1-infected CD4^+^ T cells as the first responder in HAM/TSP patients. When considering the therapeutic strategy for HAM/TSP patients, the regimen, which has an inhibitory activity on the transmigration of HTLV-1-infected cells into the spinal cord, might be recommended as one of the therapeutic strategies. At this point, integrin-targeting drugs, such as natalizumab or vedolizumab, which function by the inhibition of ligand to integrins and are already utilized as the treatment against multiple sclerosis or inflammatory bowel diseases [85], might have the potential to be one of the candidates in the therapeutic strategy for HAM/TSP patients. In addition, many kinds of inhibitors for Cdc42 signaling, which are currently under development [86], might become available in the treatment of HAM/TSP patients in the future. However, at present, based on the efficacy of PPS treatment in an open-labeled design, further studies are warranted for the evaluation of PPS as a treatment against HAM/TSP in a large-scale, randomized, controlled study.

## Figures and Tables

**Figure 1 pathogens-12-00492-f001:**
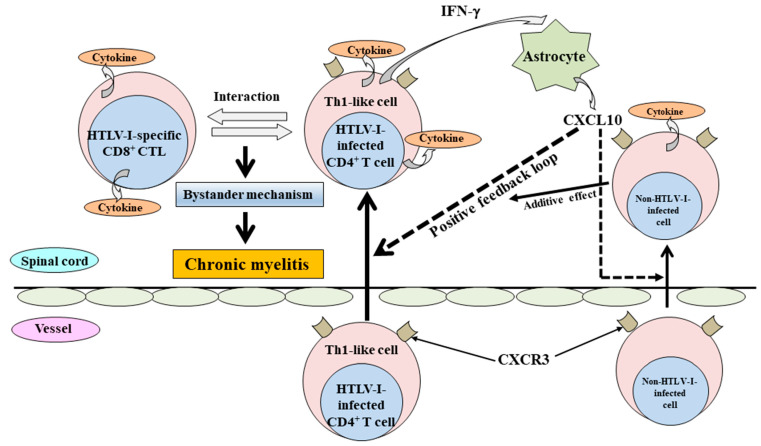
The proposed mechanism for the induction of chronic myelitis in HAM/TSP. In the induction of chronic myelitis, a long-standing bystander mechanism, such as the destruction of surrounding tissues by inflammatory cytokines, etc., induced under the interaction between infiltrated HTLV-1-infected Th1-like CD4^+^ T cells and HTLV-1-specific CD8^+^ cytotoxic T cells in the spinal cord, is probably critical. For the bystander mechanism, a positive feedback loop formed through chemokine CXCL10 (a ligand of CXCR3) from astrocytes via stimulation by IFN-γ produced from infiltrated HTLV-1-infected Th1-like CD4^+^ T cells might be involved in the maintenance and promotion of chronic myelitis. In addition, non-HTLV-1-infected CXCR3^+^ cells, which are attracted by CXCL10, provide the additive effect for a positive feedback loop.

**Figure 2 pathogens-12-00492-f002:**
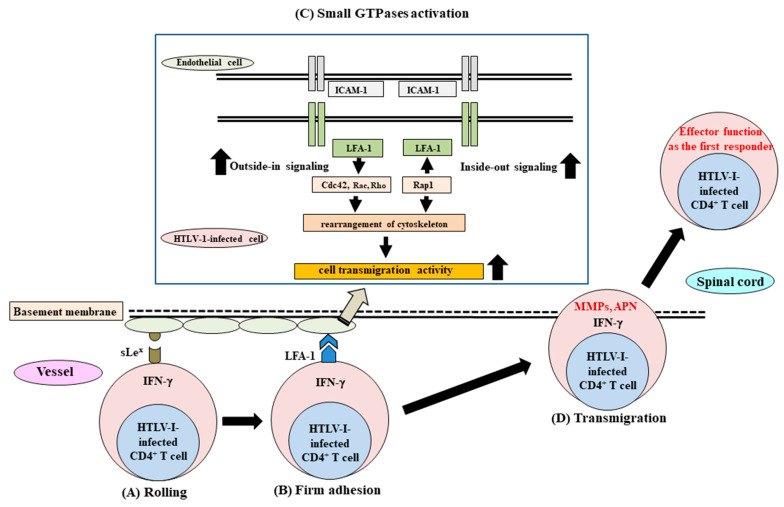
The series of molecular events involved in the heightened transmigrating activity of HTLV-1-infected CD4^+^ T cells in the peripheral blood of HAM/TSP patients into the tissues. (**A**) Rolling stage: Activation of this stage is induced by the increased number of sLe^x+^HTLV-1-infected CD4^+^ T cells in which IFN-γ production is activated. (**B**) Firm adhesion stage: Following the rolling stage, firm adhesion is induced by heightened expression of integrins, including LFA-1, on HTLV-1-infected CD4^+^ T cells. (**C**) Small GTPase activation: Following the firm adhesion stage, small GTPases function as the propulsion for the transmigration of the cells through the rearrangement of the cytoskeleton. In the square box, the up-regulated status of both the outside-in signaling and the inside-out signaling from integrin signaling in the HTLV-1-infected CD4^+^ T cells are presented. Activation of both outside-in integrin signaling and inside-out integrin signaling induced by the activation of small GTPases, such as Cdc 42 and Rap1, respectively, might upregulate polarization and cytoskeletal rearrangement, leading to an enhancement of cell transmigration activity into the tissues. (**D**) Transmigration stage: the disruption of the basement membrane is induced by cell surface proteases, including MMPs and APN produced from HTLV-1-infected CD4^+^ cells concomitant with IFN-γ production; subsequently, HTLV-1-infected CD4^+^ T cells transmigrate into the spinal cord. sLe^x^; sialyl Lewis^x^ antigen, LFA-1; lymphocyte function antigen-1, ICAM-1; intercellular adhesion molecule-1, MMPs; matrix metalloproteinases, APN; aminopeptidase-N.

## Data Availability

Not applicable.

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
