# Peer review of "HAM/TSP Pathogenesis: The Transmigration Activity of HTLV-1-Infected T Cells into Tissues"

_pathogens, 2023, doi:10.3390/pathogens12030492_

Round 1

Reviewer 1 Report

Title: HAM/TSP pathogenesis: The transmigration activity of 2 HTLV-1-infected T cells into tissues

Lines 38-40

“Of them, a Th1-like status 38 based on a high HTLV-I proviral load in the peripheral blood may be very critical in the immunopathogenesis of HAM/TSP”

A: I suggest that you discuss this matter further. This statement is important for the introduction, perhaps describe in a separate paragraph or place in the benning of the page.

Lines 92-94 

“This was previously reported in a study of the peripheral blood 92 CD4+ T cells of HAM/TSP patients; the frequency of sLex+ cells (in which Th1 cytokine, 93 such as IFN-γ, production is activated) was increased and the HTLV-I provirus was 94 concentrated in the sLex+ cell population”

A: Describe more about the study, such as the type of analysis performed, the particularities of the analyzed group and the sample size .

Lines 98-100 

“Following the rolling process on ECs, the firm adhesion of T cells by the interaction between integrins and their ligand play a crucial role as the preliminary step for transmigration of T cell into the tissue”

A: It sounds repetitive. Could be added to the above text? (Lines 89-91)

Introduction and Review.

A: Several studies were cited, it is very important to always the small sample size of this study, as well as possible particularities of this group (if any) and methodology used. Even briefly, to facilitate possible parameterizations.

____________________________________________________________________________

Suggestions

Abstract

It should develop a methodology for your abstract. It is important for the reader to obtain structural aspects, such as the quantity and relevance of the articles used for each objective. This can highlight possible gaps. In addition, some review use artificial intelligence platforms to locate related articles.

Methodology

Your article is clear, well-structured and consistent with the issues discussed. Even so, I believe that the addition of some tables summarizing the findings on the cellular alterations responsible for the increased expression of adhesion molecules. Furthermore, the function and activation of small GTPases, would facilitate a better understanding of these studies.

Author Response

Thank you very much for your valuable comments. According to your comments, I revised my manuscript.

  • Lines 38-40

“Of them, a Th1-like status 38 based on a high HTLV-I proviral load in the peripheral blood may be very critical in the immunopathogenesis of HAM/TSP”

A: I suggest that you discuss this matter further. This statement is important for the introduction, perhaps describe in a separate paragraph or place in the benning of the page.

(Comments)

According to your comment, I added the sentences as described in lines 39-50 by blue color. In addition, I added 2 references (Ref. 8 and 10) by blue color.

  • Lines 92-94

“This was previously reported in a study of the peripheral blood 92 CD4+ T cells of HAM/TSP patients; the frequency of sLex+ cells (in which Th1 cytokine, 93 such as IFN-γ, production is activated) was increased and the HTLV-I provirus was 94 concentrated in the sLex+ cell population”

A: Describe more about the study, such as the type of analysis performed, the particularities of the analyzed group and the sample size .

(Comments)

According to your comment, I described more sentences in lines 109-117 as indicated by blue color.

  • Lines 98-100

“Following the rolling process on ECs, the firm adhesion of T cells by the interaction between integrins and their ligand play a crucial role as the preliminary step for transmigration of T cell into the tissue”

A: It sounds repetitive. Could be added to the above text? (Lines 89-91)

(Comments)

I am sorry. I think that these sentences seem not to be repetitive. Therefore, I did not make the changes for them.

  • Introduction and Review.

A: Several studies were cited, it is very important to always the small sample size of this study, as well as possible particularities of this group (if any) and methodology used. Even briefly, to facilitate possible parameterizations.

(Comments)

I am very sorry. I did not have no idea how I should revise the manuscript about this point.

  • Suggestions

Abstract

It should develop a methodology for your abstract. It is important for the reader to obtain structural aspects, such as the quantity and relevance of the articles used for each objective. This can highlight possible gaps. In addition, some review use artificial intelligence platforms to locate related articles.

(Comments)

I am very sorry. I did not have no idea how I should revise the manuscript about this point.

Methodology

Your article is clear, well-structured and consistent with the issues discussed. Even so, I believe that the addition of some tables summarizing the findings on the cellular alterations responsible for the increased expression of adhesion molecules. Furthermore, the function and activation of small GTPases, would facilitate a better understanding of these studies.

(Comments)

I understand your comments. However, as I described these findings in Fig. 2 and its legend as the cellular alterations about the changes of adhesion molecules expression, etc., I did not make the table about these findings.

Reviewer 2 Report

Authors have looked into the infiltration or transmigration of T cells into tissue.

1- It is unclear whether they consider only the infected cells as in the title, or not. Indeed the feedback presented in figure 1 could favor the recruitment of other leukocytes.

2- Authors should insist more on the disruption of a particuliar barrier: the BBB. Indeed, they do not explain the importance of cytokines or infection on this mechanism.

3- Authors seem to focuss on the infiltration through endothelia. They should also present the possibiliy of infiltration across epithelia (ex. Choroid plexus, passage through the mammary gland epithelium as infected lymphocytes are present in the milk)

4- authors may have missed the infiltration in the context of dermatitis

Author Response

Thank you very much for your valuable comments. According to your comments, I revised my manuscript.

1- It is unclear whether they consider only the infected cells as in the title, or not. Indeed the feedback presented in figure 1 could favor the recruitment of other leukocytes.

(Comments)

Yes, your comment might be right. The recruitment of other leukocytes except HTLV-1-infected cells might also play an important role for the development of chronic myelitis. However, at present, it is unclear how or how much these cell population is involved in the development of chronic myelitis. Therefore, I focused the discussion about the characters of HTLV-1-infected cells as the first responder in this review.

2- Authors should insist more on the disruption of a particular barrier: the BBB. Indeed, they do not explain the importance of cytokines or infection on this mechanism.

(Comments)

This point is also important. Indeed, BBB is consisted of three kinds of cells (endothelial cells with the tight junction, pericytes, and astrocytes) and the basement membrane. Although I described “Romero et al. demonstrated that the enhanced adhesion of HTLV-1-infected cells to and the transmigration of HTLV-1-infecetd cells through the brain ECs is followed by increased brain endothelial permeability along increased expression of LFA-1 [32]. This suggests that the enhanced adhesion activity of HTLV-1-infecetd cells to ECs itself induces, not only the increased transmigration of HTLV-1-infecetd cells into the tissues but also the facilitation of the disruption of blood-brain barrier (BBB), leading to the influx of harmful agents to the tissues. The disruption of BBB, such as the change in tight junctions between ECs was observed in the spinal cord of HAM/TSP patients [33].” in lines 129-137, I added the sentence “This BBB breakdown, which is supposed to be induced by HTLV-1 infection to ECs [34] or proinflammatory cytokine secretion [33] following to the increased adhesion to ECs of HTLV-1-infected cells, might strongly contribute to the acceleration of the neuronal damages induced by bystander mechanism.” by red color in line 137-141.

And I added one reference (Ref. 34) by red color.

3- Authors seem to focuss on the infiltration through endothelia.They should also present the possibiliy of infiltration across epithelia (ex. Choroid plexus, passage through the mammary gland epithelium as infected lymphocytes are present in the milk)

(Comments)

Indeed, the study with regard to the transmigration of HTLV-1-infected cells through epithelia is very important. However, I focused the study of the transmigration of HTLV-1-infected cells through endothelia from the viewpoint of the transmigration of HTLV-1-infected cells to the systemic tissues in this review.

4- authors may have missed the infiltration in the context of dermatitis

(Comments)

I added one reference (Ref. 71) by red color.

Reviewer 3 Report

This review discusses the pathogenesis of HAM/TSP and related inflammatory diseases caused by HTLV-1 infection, with a particular focus on the migration mechanism of CD4+ T cells infected by HTLV-1. Overall, the article is very well organized and should be of interest and informative to relevant researchers. My comments are as follows:

1. Please discuss if the selective expression of various adhesion molecules is associated with Tax and HBZ expressing in HTLV-1-infected CD4+ T cells. The molecular mechanism by which the adhesion molecules are selectively expressed in HAM/TSP is critical to consider novel therapeutic of HAM/TSP in the future.

2. Please discuss more in detail the potential contribution of HTLV-1-specific CD8+ T cells in regulating and/or exacerbating HAM/TSP and related inflammatory diseases.

3. I noticed many deletion lines remaining in the text; the author should carefully check English in the text.

Author Response

Thank you very much for your valuable comments. According to your comments, I revised my manuscript.

  1. Please discuss if the selective expression of various adhesion molecules is associated with Tax and HBZ expressing in HTLV-1-infected CD4+ T cells. The molecular mechanism by which the adhesion molecules are selectively expressed in HAM/TSP is critical to consider novel therapeutic of HAM/TSP in the future.

(Comments)

Yes. This point is an very important question. However, although it was already reported that the expression of LFA-3, ICAM-1, VCAM-1, and ALCAM, etc., are related to HTLV-1 tax or HBZ activity, I could not, unfortunately, find the literatures about adhesion molecules selectively expressed in HAM/TSP.

  1. Please discuss more in detail the potential contribution ofHTLV-1-specific CD8+ T cells in regulating and/or exacerbating HAM/TSP and related inflammatory diseases.

(Comments)

According to your comment, I added the sentences as described in lines 60-65 by green color. In addition, I added 2 references (Ref. 14 and 15) by green color.

  1. I noticed many deletion lines remaining in the text; the author should carefully check English in the text.

(Comments)

I deleted the deletion lines in my manuscript.